# pH as a Key Factor for the Quality Assurance of the Preparation of *Gastrodiae Rhizoma* Formula Granules

**DOI:** 10.3390/molecules27228091

**Published:** 2022-11-21

**Authors:** Shuting Xie, Ke Min, Hai Li, Ying Wang, Mincong Liu, Ming Ma, Desheng Zhou, Haijun Tu, Bo Chen

**Affiliations:** 1Key Laboratory of Phytochemical R&D of Hunan Province, Key Laboratory of Chemical Biology & Traditional Chinese Medicine Research of Ministry of Education, College of Chemistry and Chemical Engineering, Hunan Normal University, Changsha 410081, China; 2The First Affiliated Hospital of Hunan University of Chinese Medicine, Changsha 410007, China; 3College of Biology, Hunan University, Changsha 410082, China

**Keywords:** *Gastrodiae rhizoma*, gastrodin, parishins, pH, HPLC

## Abstract

*Gastrodiae rhizoma* (*GR*) formula granules and preparations have been used as a popular traditional Chinese medicine for clinical treatment since they have good pharmacological activity to treat nervous system diseases. Gastrodin and parishins have been the main active components in aqueous extracts for *GR* formula granules, but their pharmacological activities and metabolism are different. For quality control of the extracts, the extraction conditions should be investigated to accurately control the contents of two kinds of components. In this paper, the transfer rate of six index components (including gastrodin, *p*-hydroxybenzyl alcohol, parishin A, parishin B, parishin C, and parishin E) obtained by HPLC were used as indicators to investigate the effect of pH on the *GR* extraction process. The results demonstrated that pH is a key factor for preventing transforming parishins into gastrodin and maintaining high content of parishins in the extracts. It can be concluded that the weak acid environment could improve the transfer rate of parishins, thus ensuring the gastrodin and parishins consistency between *GR* raw materials and its aqueous extracts. Therefore, pH is an essential condition for accurate quality control of the extracts.

## 1. Introduction

*Gastrodiae rhizoma* (*GR*) is the dried tuber of Orchidaceae, which has the effects of calming wind and spasmolysis, calming liver yang, expelling wind, and dredging collaterals [1,2]. It is widely utilized in infantile convulsion, epilepsy, convulsion, headache, vertigo, etc. [3,4,5,6,7,8]. It is stipulated that the characteristic map of *GR* formula granules in National Medical Products Administration was six index components, including gastrodin, *p*-hydroxybenzyl alcohol, parishin A, parishin B, parishin C, and parishin E [9]. Pharmacological studies showed that gastrodin exhibits neuro-protective effects against cerebral I/R injury [10]. *p*-Hydroxybenzyl alcohol can provide neuroprotection by preventing brain damage [11,12]. Parishins, such as parishin A, parishin B, parishin C, and parishin E, had significant protective effects on myocardial cells [13,14,15,16]. In summary, *GR* formula granules and preparations have been used as the popular traditional Chinese medicine for clinical treatment since their effective pharmacological activity. It should be noted that gastrodin has a short action time in vivo, so it needs multiple doses to be treated clinically. Fortunately, parishins significantly prolong t1/2 (P0.01), which is 2.7-fold time of gastrodin, respectively [17]. However, it is common to pay more attention to the content of gastrodin in *GR* formula granules than parishins, resulting in different levels of parishins in *GR* formula granules in the market, which is against standardized quality control of *GR* formula granules as well as reducing the pharmacological activity of *GR*. Hence, investigating the content of multiple index components containing parishins in *GR* formula granules for improving the retention of parishins in *GR* formula granules is important.

Previous studies showed the presence of a large amount of parishins in *GR* formula granules [18,19,20,21]. Parishins were conjugated consisting of one citric acid unit and multiple gastrodins units by ester linkages. Parishins contain a large number of ester groups that are easy to hydrolyze, and can be hydrolyzed under too acidic or too alkaline conditions to produce gastrodin, *p*-hydroxybenzyl alcohol, and a large number of byproducts, which may lead to the loss of parishins in *GR* preparation such as *GR* formula granules, Huoxue Rongluo Pills and Tianma Gouteng Decoction, further resulting in a decline in efficacy [18]. Thus, controlling the pH of the preparation process of *GR* formula granules can further improve the preparation technology standard of *GR* formula granules and better ensure that the formula granules purchased in the market have the same medicine effect as the traditional Chinese medicine decoction pieces. In recent years, alkali hydrolysis-HPLC is usually applied to investigate the effect of alkali concentration on hydrolysis [22]. Yet, these studies were only focused on determining the content of free and total gastrodin in *GR* formula granules, and pay little attention to the effect of pH. However, pH was generally a key factor in quality control of formula granules; therefore, investigating the appropriate range of pH in the preparation of *GR* formula granules is meaningful for its standardization.

As a simple, rapid, and accurate method, HPLC is often used in quantitative analysis of components in traditional Chinese medicine [23]. Based on this, a method of alkali hydrolysis-HPLC for quantitative analysis of gastrodin, *p*-hydroxybenzyl alcohol, parishin A, parishin B, parishin C, and parishin E from *GR* formula granules and preparations was established. Taking parishin A as a model compound of parishins, the stability of parishin A in different pH environments was investigated. In addition, we prepared 14 batches of *GR* standard decoctions in weak acid environment, and detected the transfer rates of the above six index components by HPLC, taking them as indexes to estimate the retention rate of parishins at the reasonable pH. The simplified workflow is shown in Figure 1. In this work, we measured the content of six index components from 14 batches of *GR* decoction pieces and standard decoctions, and investigated the effect of pH on the retention rate of six index components in the preparation process of *GR* formula granules. The results demonstrated effective improvement of the retention of parishins in *GR* formula granules and related preparations by controlling the solution in weak acid environment, greatly improved the efficacy of *GR* formula granules, and provides a reference for the quality control of preparations.

## 2. Results and Discussion

### 2.1. Quantitative Analysis of Index Components in GR by HPLC

To accurately determine the index components such as gastrodin, *p*-hydroxybenzyl alcohol, parishin A, parishin B, parishin C, and parishin E, the sample preparation conditions were optimized and validation of the method was completed.

The extraction efficiency of index components from *GR* with different solvents such as ethanol, MeOH, water, 25% MeOH, 50% MeOH, and 75% MeOH was investigated. The extraction efficiency of the components was the highest in 50% MeOH solution (see Figure 2a). In addition, ultrasonic for 30 min and the 1/50 ratio of material to liquid could completely extract six index components from *GR* (see Figure 2b,c).

The selection of the HPLC conditions was guided with the requirement for obtaining chromatograms with satisfactory resolutions of the analytes. To achieve more chemical information and effective separation of each component, taking 50% MeOH as blank solvent, and *GR* decoction piece (No. S2), *GR* standard decoction (No. Y2), and commercially available *GR* formula granule as test solutions, and mixed standard solution as references, the separation of six index components reached the requirements of quantitative analysis by optimizing the gradient of HPLC (see Figure 3). As shown in Figure 3, these peaks from six index components are presented in the same retention time of *GR* decoction piece, *GR* standard decoction, commercially available *GR* formula granules, and the mix standard solution, but no corresponding peaks in the blank solvent, which indicates that other impurities in the test solutions have no interference with the determination of the above six index components. In addition, we also investigated the parameter of peak symmetry and peak capacity in gradient elution. As listed in Table 1, the tailing factor (T) of six index compounds were between 0.848 and 1.045, which meets the peak symmetry requirements. This 250 mm chromatographic column can reach the peak capacity of 43 under the optimized conditions [24].

The method validation was conducted according to the accepted Guidance for Analytical Method Validation in Chinese Pharmacopoeia (2020 edition). In order to verify the feasibility of the quantitative method, we first constructed calibration curves for six index components by HPLC. According to the calibration curve as Table 1, the limit of detection (LOD, S/N = 3) of gastrodin, *p*-hydroxybenzyl alcohol, parishin A, parishin B, parishin C, and parishin E were 0.040, 0.035, 0.050, 0.050, 0.057, and 0.500 mg/L, and the limit of quantitation (LOQ, S/N = 10) were 0.130, 0.090, 0.150, 0.150, 0.185, and 1.650 mg/L, respectively. It was found that all compounds had excellent linearity (all *R*^2^ were greater than 0.9996). In order to prove that the method produces reliable results, we evaluated the uncertainty of slope and intercept in the linear equation. The results listed in Table 1 show that the uncertainties of slope and intercept in the six standard curves are within the expected range, which can provide support for data validity [25].

Second, to evaluate the performance of this method, we also investigated its recovery, precision, repeatability, and stability. As is shown in Appendix A, the recovery rates range from 96.5% to 103.0%, which ensured the accuracy of the sample determination. In addition, the precision for the six analytes was also acceptable with the RSD of 0.20–1.45%. The results of stability test demonstrated that RSD values of the peak areas for the six analytes were less than 1.53%. In addition, the RSD of 1.03–2.75% in parallel experiments proved the method was completely repeatable. The above validated items mentioned above verified that HPLC method established was appropriate for further analysis of *GR* samples.

#### Determination of Six Index Components in GR Real Samples

In this study, the content of gastrodin, *p*-hydroxybenzyl alcohol, parishin A, parishin B, parishin C, and parishin E in *GR* decoction pieces and standard decoctions was determined by HPLC. A total of 14 batches of *GR* decoction pieces and standard decoctions were prepared in parallel, and the content of the six index components in each batch were calculated, and these results were listed in Table 2. We found that the content of six index components of *GR* decoction pieces from different sources were quite varied, which may be due to different planting environments in producing areas and processing techniques of medicinal materials (see Table 2). Furthermore, the total contents of gastrodin and *p*-hydroxybenzyl alcohol in 14 batches of standard decoctions were all higher than 0.25% to a great extent, which meets the standard of “Pharmacopoeia of China”. Overall, the above experiments demonstrated that proposed HPLC is not only capable of simultaneous analysis of six index components in *GR*, but also possesses good method performances, which provides a facilitating tool for quantitative analysis of main components in traditional Chinese medicine.

### 2.2. The Effect of pH on the Transfer Rate of Parishins in GR Formula Granules

Parishins in *GR* formula granules may hydrolyze if the pH of the solution is too acidic or too alkaline due to the instability of ester group, resulting in the loss of pharmacological activity. In addition, there are different preparation processes of major pharmaceutical companies, the content of parishins in *GR* formula granules on the market is uneven, and the efficacy is different. Therefore, it is urgent to investigate the appropriate range of pH and standardize the preparation technology of *GR* formula granules for the quality assurance of the preparation of *GR* formula granules. Parishin A, as a typical model compound of parishins, is a glycoside compound formed by dehydration and condensation of one molecule of citric acid and three molecules of gastrodin. Under certain conditions, it can undergo a hydrolysis reaction to produce gastrodin monomer and *p*-hydroxybenzyl alcohol, and its potential hydrolysis path was shown in Appendix A, and its hydrolysis in alkaline solution was successfully verified by HPLC (see Appendix A). In addition, the hydrolysis behaviors of parishin A were investigated at different pH, reaction temperature, and reaction time. We found that changing the water bath temperature and hydrolysis time will not cause the hydrolysis of parishin A under pH is 4.3, e.g., weak acid environment (the results are not given in this study), while the degree of hydrolysis of parishin A gradually increased when the pH was greater than 7.0 (Figure 4), which suggests parishin A are relatively stable under weak acidic conditions.

To this end, we prepared 14 batches of *GR* standard decoction in a weak acid environment, and calculated the paste yield of *GR* standard decoction. The paste yields of 14 batches of *GR* standard decoction ranged from 11.71% to 28.84% (see Table 3). From the results of the transfer rate in Table 3, the pH of the decoction was controlled between 3 and 6 by dripping acetic acid (e.g., weak acid environment), which ensured the relative stability of parishins and kept them in the standard decoction as much as possible, thus the hydrolysis of parishins were greatly reduced.

As shown in Figure 5, the content of hydrolysis products, such as gastrodin and *p*-hydroxybenzyl alcohol, were relatively constant, which indicates parishins did not hydrolyze in a weak acid environment (e.g., pH = 3 to 6). In addition, the transfer rate of six index components changes in the similar proportion (see Figure 5), which suggests the retention of parishins in *GR* formula granules can be improved by adjusting pH to weak acidity, thus enhancing the curative effect of *GR* formula granules. In short, the transfer rate of parishins in the preparation process of *GR* formula granules will greatly improve if the solution is controlled in weak acidity, which suggests that pH plays an important role for the quality assurance of the preparation of *GR* formula granules.

### 2.3. Importance of pH in the Preparation of GR Formula Granules

In this study, the HPLC fingerprints of 3 batches of *GR* formula granules (H1, H2, and H3) from three different hospitals in Changsha were monitored, and compared with *GR* standard decoction (Figure 6). It can be clearly seen that parishins in 3 batches of *GR* formula granules sold from hospitals were obviously lost (e.g., peak 3 to 6 decreased) compared with *GR* standard decoction, while the content of gastrodin increased significantly (e.g., peak 1 increased). This indicates that parishins, including parishin A, parishin B, parishin C, and parishin E, were mostly hydrolyzed, which may be due to the preparation of *GR* formula granules without controlling pH.

In order to estimate the effect of pH on commercially available *GR* preparations, a compound preparation containing *GR*, e.g., HuoxueRongluo Pill II, was selected as a research object. HuoxueRongluo Pill II is an in-hospital preparation of the First Affiliated Hospital of Hunan University of Traditional Chinese Medicine. It was a Chinese patent medicine for the treatment of ischemic stroke after compatibility adjustment according to the mechanism of invigorating qi deficiency and stagnation of disease in traditional Chinese medicine. It was made up of eight Chinese medicines, namely *Lonicerae japonicae caulis*, *Picriae herba*, *GR*, *Chaenomelis fructus*, *Acori tatarinowii rhizoma*, *Persicae semen*, *Angelicae dahuricae radix*, and *Paeoniae radix rubra*, according to the ratio of 6:3:3:3:2:2:2:2 without additional additives. Since parishins are effective medicinal ingredients in Huoxue Rongluo Pill II, its retention in the preparation process should be considered. To this end, the retention of parishins in Huoxuerongluo Pill II was monitored by HPLC. We can see from Appendix A that the transfer rate of parishins in Huoxuerongluo Pill II was high (approximate 80%) and there was no obvious loss. We speculated that pH is the key factor, because the water extracts of eight Chinese medicinal materials in Huoxuerongluo Pill II were all weakly acidic (see Appendix A), which makes parishins keep a stable environment, and thus a high transfer rate. Overall, we again confirmed that pH was a key factor for the quality assurance of *GR* preparations. We therefore suggest that paying more attention to the pH in the preparation process of formula granules or compound preparations, in order to avoid the situation of large loss of active ingredients.

Traditional Chinese medicine granules are granules made from single Chinese medicine decoction pieces by heating, extracting, separating, concentrating, drying, and granulating [26,27]. Formula granules overcome the problems of traditional Chinese medicine, such as difficult carrying, inconvenient preservation, and time-consuming decoction, and have a broad application prospect. Standard decoction as the intermediate of formula granules can be used as a material standard to measure the consistency between formula granules and traditional Chinese medicine decoction pieces [28]. Hence, *GR* decoction pieces, *GR* standard decoctions, and commercially available *GR* formula granules were selected as the research object in this study, their similarities and differences were investigated by HPLC.

With the increasing people studying the chemical constituents of *GR*, a new class of active constituents has been identified, including parishins. After administration, these ingredients are usually used as prodrugs to release gastrodin in the small intestine, and its half-life is 2.7 times that of gastrodin, which takes a long time in vivo. However, as the earliest active ingredient found in *GR*, gastrodin is often more easily noticed by people, but parishins, which have better efficacy, are ignored instead. In this study, we proposed that pH is a key factor for the quality assurance of the preparation of *GR* formula granules for the first time. We demonstrated that parishins will be incompletely hydrolyzed when *GR* decoction pieces were heated and extracted without considering pH condition, thus the content of gastrodin in standard decoction will be greatly increased, while parishins will not be retained. Only by strictly controlling the pH in the preparation process of *GR* formula granules to maintain weak acidity will the transfer rate of parishins be obviously improved. Considering the sensitivity of parishins in *GR* formula granules to pH, sufficient reasons confirm that the pH value is the key factor to ensure the preparation quality of *GR* formula granules.

## 3. Materials and Methods

### 3.1. Materials, Reagents, and Chemicals

Huoxuerongluo Pill II was provided by the First Affiliated Hospital of Hunan University of Chinese Medicine. *GR* formula granules were purchased from a local drugstore (Changsha, China). The samples of *GR* decoction piece were from different provinces, including Hubei (No. S1–S3), Hunan (No. S4–S6), Sichuan (No. S7–S8), Jilin (No. S9–S11), and Yunnan (No. S12–S14) (the detailed information is listed in Appendix A), which met the technical requirements for quality control and standard formulation of traditional Chinese medicine formula granules. These decoction pieces were further verified as *GR* by Professor Desheng Zhou from the First Affiliated Hospital of Hunan University of Traditional Chinese Medicine. Gastrodin, *p*-hydroxybenzyl alcohol, parishin A, parishin B, parishin C, and parishin E standards were all purchased from Chengdu Kloma Biotechnology Co., Ltd. (Chengdu, China) and their chemical structures are shown in Appendix A. Methanol and acetonitrile were both purchased from Beijing Inoke Technology Co., Ltd. (Beijing, China) Phosphoric acid, potassium dihydrogen phosphate, dipotassium hydrogen phosphate, and sodium hydroxide were all purchased from Sinopharm. The experimental water was made by SZ-93 automatic double pure water distiller.

### 3.2. Instruments

The instruments included Shimadzu LC-20A (Shimadzu Co., Ltd., Guangzhou, China), FW-80 high-speed universal pulverizer (Beijing Yongguang Medical Instrument Co., Ltd., Beijing, China), AP135W electronic balance (Shimadzu Co., Ltd., Guangzhou, China), MTN-2800D nitrogen blowing concentration device (Tianjin Aote sainz Instrument Co., Ltd., Tianjing, China), F-030SD ultrasonic cleaner (Shenzhen Fuyang Technology Group Co., Ltd., Shenzhen, China), Sorvall 16R high-speed centrifuge (Shanghai Thermo Fisher Scientific Shier Technology Co., Ltd., Shanghai, China), SZ-93 automatic double water distiller (Shanghai Yarong Biochemical Instrument Factory, Shanghai, China), and SCIENTZ-10ND freeze dryer (Zhejiang Ningbo Xinzhi Biotechnology Co., Ltd., Ningbo, China).

### 3.3. HPLC Analysis

#### 3.3.1. Preparation of Standard and Sample Solutions

Measures of 10.0 mg of gastrodin, *p*-hydroxybenzyl alcohol, parishin A, parishin B, parishin C, and parishin E standard were dissolved in 50% methanol as standard solution with the concentration of 1000 µg/mL. Then, 2.12 g KH_2_PO_4_ and 5.56 g K_2_HPO_4_ were mixed in 200 mL of distilled water to prepare 0.2 mol/L phosphate buffer (PBS) with pH 7.0, and stored at 4 °C, which was used for subsequent pH adjustment of the solution. Furthermore, 0.2 g of *GR* decoction piece powders of No. S1–S14 was ultrasonically extracted (power 200 W and frequency 53 kHz) for 30 min with 5 mL of 50% methanol. The filtered supernatant was analyzed by HPLC.

To estimate the transfer rates of six index components from decoction piece to standard decoction of *GR*, 14 batches of *GR* standard decoctions were prepared. The preparation process was as follows: 10.0 g of *GR* decoction pieces of No. S1–S14 was soaked for 30 min by appropriate amounts of distilled water, acetic acid solution was then added to keep the pH between 3 and 6. After slightly boiling for 30 min, the supernatant was filtered. The extraction procedure was repeated again. Two filtrates were combined, then freeze-dried. Then, 0.2 g of *GR* dry extract of No. Y1–Y14 was ultrasonically extracted (power 200 W and frequency 53 kHz) for 30 min with 5 mL of 50% methanol. The filtered supernatant was analyzed by HPLC.

#### 3.3.2. HPLC Analysis

The chromatographic separation was completed on an Inertsil ODS-3 column (250 mm × 4.6 mm, 5 µm), and the gradient elution was carried out with acetonitrile (B) and 0.1% phosphoric acid water solution (A). The gradient elution program was 0–5 min, 8% B, and 5–35 min, 8–40% B. The flow rate was 1.0 mL/min, the detection wavelength was 220 nm, the injection volume was 10 µL, and column temperature was kept at 30 ℃ during gradient elution.

### 3.4. Calculation of the Transfer Rate of Paste Extraction

Accurately weigh the quality of the extract, and calculate the paste rate by the formula of (the quality of dry extract/the total amount of decoction pieces) × 100%, while the transfer rate is calculated according to the ratio of the content of active ingredients in dry extract obtained by making standard decoction pieces per gram to the content of active ingredients in each gram of decoction pieces, that is, the transfer rate = *wv*/(*WvM*) [29,30], where “*w*” represents the mass concentration of ingredients in standard decoction, “*v*” represents the volume of standard decoction, “*W*” represents the mass concentration of crude drugs in standard decoction, and “*M*” represents the mass fraction of ingredients in decoction pieces.

### 3.5. Investigation on Hydrolysis Conditions of Parishin A

Parishins contain multiple ester group, they can be hydrolyzed at acid or alkaline environment. In order to obtain the optimal pH range to prevent parishins from hydrolizing, the effect of pH on hydrolysis of parishin A (a model compound of parishins) was investigated. First, 200 µL of PBS at pH 2.64, 3.86, 4.68, 5.64, 6.79, 7.64, 8.66, 9.61, and 10.30 adjusted with phosphoric acid or sodium hydroxide solution was added into a 2 mL chromatography vial. Then, 180 µL of methanol and 20 µL of 1000 µg/mL parishin A solution were then added into a vial, and heated at 100 °C in an oven for 30 min. The obtained solution was filtered through a 0.22 µm filter membrane for further quantitation analysis by HPLC. For protection against the column, the pH range of this study only examined 2.64–10.30. In parts less than 2.64 and greater than 10.30, its instability has been shown in previous studies [2,31].

## 4. Conclusions

In the course of this study, we explored the stability of parishin A at different pH and concluded that the extraction process of parishins should be controlled in a weak acid environment (pH range from 3 to 6). The results show that the transfer rate of parishins in the preparation process of *GR* formula granules will greatly improve when the solution is controlled in weak acidity, which suggests that pH plays an important role for the quality assurance of the preparation of *GR* formula granules. We therefore suggest paying more attention to the pH in the preparation process of formula granules or compound preparations to avoid the situation of large loss of parishins, thus ensuring the efficacy consistency of *GR* preparation and raw materials. Overall, this study provides some references for the preparation of formula granules and the clinical use of traditional Chinese medicine.

## Figures and Tables

**Figure 1 molecules-27-08091-f001:**
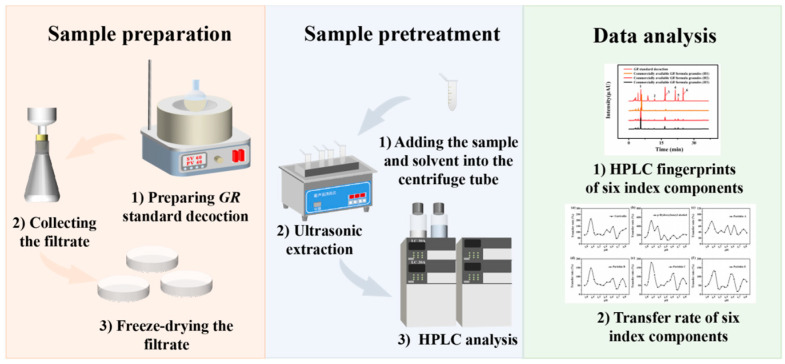
A simplified workflow illustrating the preparation, pretreatment, and detection process of *GR* formula granules.

**Figure 2 molecules-27-08091-f002:**
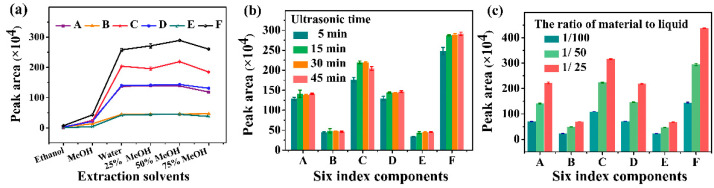
Optimization of sample preparation conditions of *GR* formula granules. (**a**) the effect of extraction solvents including ethanol, MeOH, water, 25% MeOH, 50% MeOH, and 75% MeOH, on the extraction efficiency of six index components; (**b**) the effect of ultrasonic time (5, 15, 30, and 45 min) on the extraction efficiency of six index components; and (**c**) the effect of the ratio of material to liquid (1/100, 1/50, and 1/25) on the extraction efficiency of six index components.

**Figure 3 molecules-27-08091-f003:**
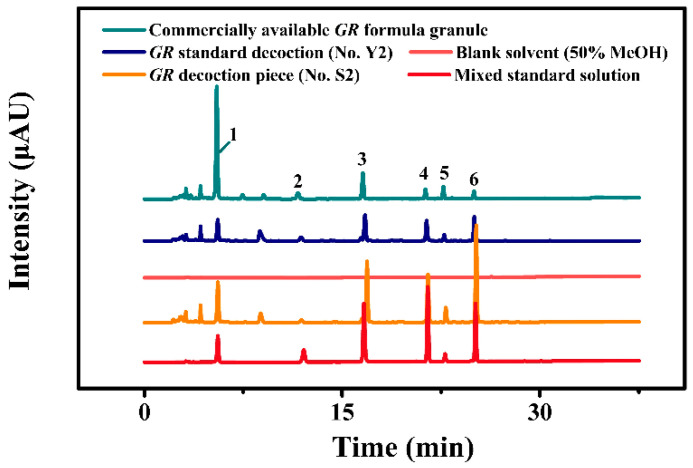
Typical HPLC chromatogram of blank solvent (50% MeOH), *GR* decoction piece (No. S2), *GR* standard decoction (No. Y2), commercially available *GR* formula granule, and mixed standard solution.

**Figure 4 molecules-27-08091-f004:**
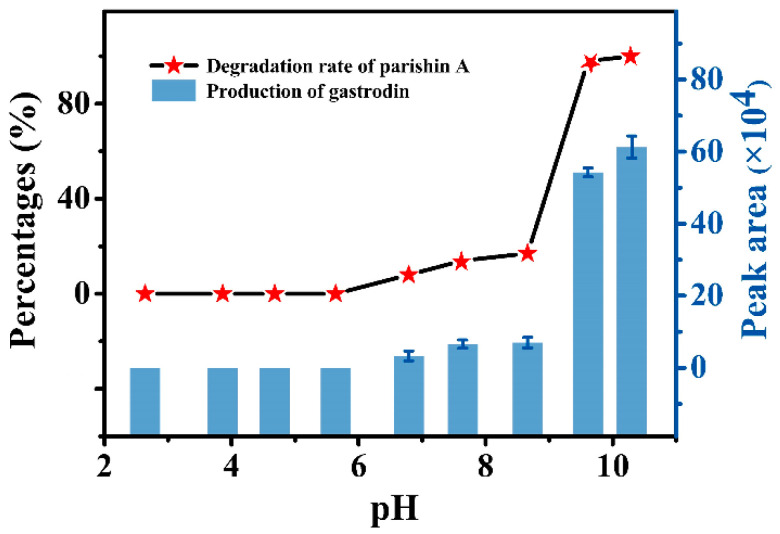
The effect of pH on hydrolysis of parishin A.

**Figure 5 molecules-27-08091-f005:**
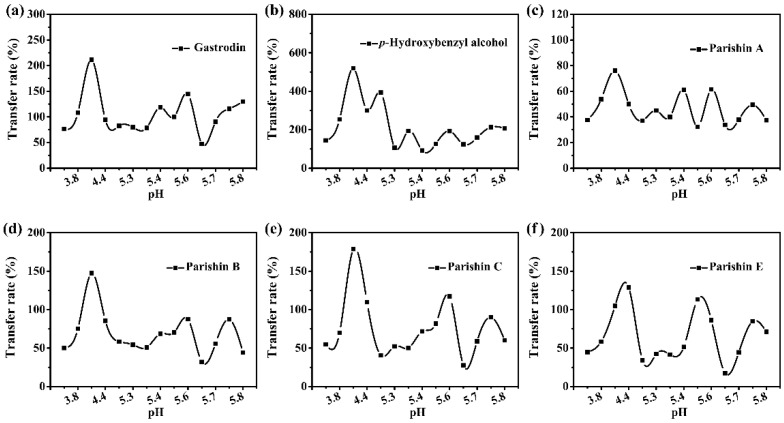
The effect of pH on the transfer rate of six index components, incuding (**a**) gastrodin, (**b**) *p*-hydroxybenzyl, (**c**) parishin A, (**d**) parishin B, (**e**) parishin C, and (**f**) parishin E in *GR* standard decoction at weak acid environment.

**Figure 6 molecules-27-08091-f006:**
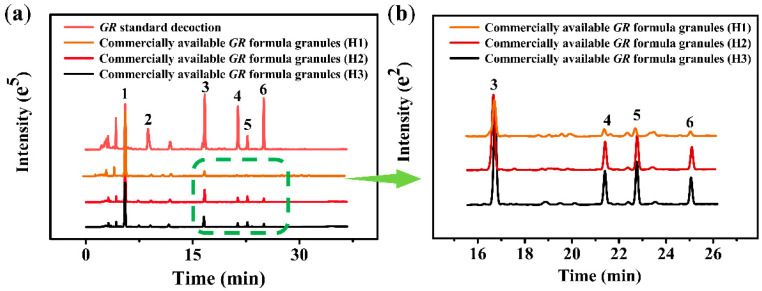
(**a**) Comparison of HPLC fingerprints between 3 batches of commercially available *GR* formula granules (H1, H2, and H3) and *GR* standard decoction. (**b**) Enlarged view of the mark areas with green lines in (**a**).

**Table 1 molecules-27-08091-t001:** In this experiment, the standard curve, limit of detection (LOD, S/N = 3), and limit of quantitative (LOQ, S/N = 10) of six index components of GR were determined by HPLC. All calibration points were repeated three times.

Compound	Calibration Curve	Uncertainties of Slopes	Uncertainties of Intercepts	Linear Range (mg/L)	*R* ^2^	LOD (mg/L)	LOQ (mg/L)	T
gastrodin	*y* = 20,322 *x* − 47,498	50	2.6 × 10^4^	20–1000	0.9999	0.040	0.130	0.928
*p*-hydroxybenzyl alcohol	*y* = 37,946 *x* − 30,919	136	6.8 × 10^4^	5–1000	0.9996	0.035	0.090	1.045
parishin A	*y* = 18,330 *x* − 36,509	98	5.7 × 10^4^	20–1200	0.9999	0.050	0.150	1.028
parishin B	*y* = 16,156 *x* − 4942.5	100	5.7 × 10^4^	20–1200	0.9999	0.050	0.150	1.039
parishin C	*y* = 13,934 *x* − 1890.9	96	5.5 × 10^4^	1–1000	0.9996	0.057	0.185	1.001
parishin E	*y* = 12,300 *x* − 12,5776	96	5.6 × 10^4^	20–1200	0.9997	0.500	1.650	0.848

**Table 2 molecules-27-08091-t002:** Contents of six components in 14 batches of *GR* decoction pieces and standard decoctions (*n* = 3).

Samples	No.	Mass Fraction (mg/g)					
		Gastrodin	*p*-Hydroxybenzyl Alcohol	Parishin A	Parishin B	Parishin C	Parishin E
*GR* decoction pieces	S1	4.1 ± 1.9	0.2 ± 1.9	7.8 ± 0.6	3.7 ± 0.7	1.7 ± 0.7	6.3 ± 0.8
	S2	1.6 ± 1.8	0.1 ± 1.1	3.6 ± 2.8	2.0 ± 1.6	0.8 ± 2.8	4.3 ± 2.1
	S3	5.0 ± 2.2	1.5 ± 0.7	8.8 ± 0.5	7.3 ± 0.6	2.6 ± 0.6	9.6 ± 1.6
	S4	4.2 ± 2.2	0.3 ± 1.6	6.3 ± 1.8	4.8 ± 2.5	1.7 ± 1.7	12.6 ± 2.2
	S5	2.9 ± 2.2	0.2 ± 2.1	4.0 ± 1.3	3.6 ± 1.2	1.1 ± 2.1	9.5 ± 0.8
	S6	1.6 ± 1.4	0.2 ± 1.4	4.1 ± 1.1	2.6 ± 1.7	0.6 ± 2.3	8.3 ± 2.1
	S7	1.4 ± 2.1	0.8 ± 0.9	6.0 ± 2.3	3.2 ± 2.1	0.8 ± 1.8	8.4 ± 1.6
	S8	3.3 ± 0.2	0.6 ± 2.0	10.2 ± 1.7	5.1 ± 0.6	1.3 ± 2.1	6.0 ± 2.3
	S9	1.8 ± 1.1	0.5 ± 1.1	1.4 ± 1.8	1.8 ± 2.0	0.4 ± 1.1	6.7 ± 1.7
	S10	2.3 ± 1.0	0.5 ± 1.7	3.8 ± 1.5	2.9 ± 0.8	0.8 ± 0.3	8.7 ± 0.7
	S11	2.5 ± 1.9	0.5 ± 0.5	6.1 ± 2.1	4.0 ± 1.4	1.2 ± 1.9	4.9 ± 2.6
	S12	1.6 ± 2.0	1.8 ± 0.7	3.4 ± 2.4	4.0 ± 0.9	0.8 ± 1.0	10.0 ± 0.7
	S13	1.6 ± 1.2	1.8 ± 1.2	4.0 ± 1.1	4.6 ± 1.2	0.8 ± 1.5	8.0 ± 1.0
	S14	1.5 ± 1.8	0.4 ± 1.5	3.3 ± 1.7	1.9 ± 0.8	0.5 ± 1.8	4.7 ± 4.1
*GR* standard decoction	Y1	20.2 ± 0.7	2.8 ± 3.4	20.9 ± 0.4	12.8 ± 0.3	5.3 ± 0.7	15.5 ± 0.9
	Y2	11.1 ± 0.8	3.4 ± 2.0	10.4 ± 2.6	9.9 ± 1.5	2.9 ± 2.8	13.5 ± 3.1
	Y3	24.4 ± 1.2	12.7 ± 2.0	20.9 ± 2.3	21.8 ± 0.5	8.3 ± 1.6	19.5 ± 3.5
	Y4	11.1 ± 3.7	2.2 ± 3.1	6.1 ± 3.2	8.6 ± 3.8	2.5 ± 3.2	23.6 ± 3.7
	Y5	17.1 ± 2.9	1.4 ± 2.6	14.2 ± 3.5	13.1 ± 2.7	5.4 ± 2.6	24.0 ± 3.5
	Y6	6.6 ± 1.3	1.7 ± 3.6	12.0 ± 3.2	7.9 ± 2.8	2.0 ± 2.8	14.2 ± 2.1
	Y7	10.5 ± 2.6	4.8 ± 0.1	19.8 ± 0.1	14.0 ± 0.1	3.5 ± 0.1	32.7 ± 0.1
	Y8	10.2 ± 1.9	3.5 ± 4.4	18.4 ± 2.9	10.3 ± 1.3	2.8 ± 1.1	9.1 ± 2.8
	Y9	12.0 ± 3.2	11.4 ± 3.2	12.9 ± 3.0	10.6 ± 2.5	2.9 ± 3.0	23.5 ± 1.2
	Y10	10.4 ± 2.9	6.1 ± 2.9	9.3 ± 2.9	8.5 ± 1.0	2.3 ± 1.3	20.1 ± 0.8
	Y11	10.2 ± 3.2	2.5 ± 2.6	13.0 ± 3.1	11.1 ± 1.2	3.3 ± 2.8	11.1 ± 1.2
	Y12	11.0 ± 3.3	15.8 ± 1.1	27.2 ± 2.5	19.6 ± 2.1	4.5 ± 1.6	22.3 ± 3.3
	Y13	12.5 ± 3.4	23.3 ± 3.7	17.4 ± 2.4	12.5 ± 2.5	3.0 ± 2.4	18.4 ± 2.6
	Y14	19.8 ± 0.7	12.6 ± 2.5	21.2 ± 1.4	17.5 ± 0.2	6.0 ± 2.0	22.0 ± 0.6

**Table 3 molecules-27-08091-t003:** The extraction rate of *GR* standard decoction (No. Y1–Y14) and transfer rate of six index components (*n* = 3).

No.	Paste Yield (%)	pH	Transfer Rate (%)					
			Gastrodin	*p*-Hydroxybenzyl Alcohol	Parishin A	Parishin B	Parishin C	Parishin E
Y1	21.9 ± 1.3	3.8	108.2	254.0	53.8	75.3	70.0	58.3
Y2	11.7 ± 2.1	4.7	82.6	393.5	37.1	58.3	40.5	34.1
Y3	18.6 ± 0.9	5.7	90.5	159.3	37.8	55.6	58.9	44.4
Y4	18.0 ± 1.0	5.6	47.5	124.0	33.7	32.0	27.6	17.3
Y5	24.4 ±1.2	5.6	144.8	192.6	61.5	87.7	117.3	86.3
Y6	28.8 ± 2.9	5.7	115.9	213.2	49.5	87.4	90.1	84.8
Y7	15.7 ± 2.2	5.4	118.7	91.9	61.1	68.7	71.8	51.6
Y8	24.9 ± 1.6	3.0	76.4	144.6	37.6	50.1	54.6	44.7
Y9	14.3 ± 2.1	4.4	94.3	300.5	50.0	85.5	109.6	128.9
Y10	17.2 ± 0.6	5.3	78.7	193.9	39.9	50.9	50.2	41.6
Y11	19.9 ± 0.3	5.3	79.8	106.3	44.9	54.4	52.3	42.4
Y12	14.4 ± 1.1	5.4	100.0	126.6	32.3	70.5	81.9	113.2
Y13	16.2 ± 1.0	5.8	130.0	206.9	37.4	44.2	60.2	71.2
Y14	16.1 ± 1.8	4.3	211.4	519.4	76.1	147.4	178.7	104.8

## Data Availability

The data presented in this study are available in this article.

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
