# Peer review of "pH as a Key Factor for the Quality Assurance of the Preparation of Gastrodiae Rhizoma Formula Granules"

_molecules, 2022, doi:10.3390/molecules27228091_

Round 1

Reviewer 1 Report

The manuscript “pH as a key factor for the quality assurance of the preparation of Gastrodiae Rhizoma formula granules” presents the results concerning the suitable conditions of preparation of a Chinese medicine product. The information could be of interest for a relatively narrow audience which limits the utility of the data.

The manuscript require some improvement before publication.

1)      English language editing is required. Some sentences are not clear.

2)      Introduction should provide information about the chemical nature of the compounds involved as well as the processes taking place at elevated pH. Some similar products should be mentioned.

3)      Fig 2 – too small font size

4)      Table 1 – uncertainties of slopes and intercepts are missing,

5)      Table 2 and Table 3 – too many significant digits were provided. Only 1 or 2 digits for uncertainties and corresponding number of digits in figures should be used.

Reviewer 2 Report

figure 1 should be improved in term of the quality of headings, the font size should be increased.

in validation, the parameter of peak symmetry and peak capacity in gradient elution should be added.

column oven temperature is missing. please add it.

figure 5 - add the numbers at left axis for better orientation

figure 6 -zoom the bottom three chromatograms.

why the hydrolysis graphs/results at alkaline medium is missing?

Round 2

Reviewer 1 Report

It seems to me that the correction provided are sufficient.

The presentation of uncertainties in Table 1 is somewhat unsatisfactory, however - an exponent should be used to reduce the number of significant digits required for presentation.

Author Response

Thanks for your valuable suggestion. We have expressed the uncertainty in Table 1 in exponential form. Please check it. Thanks.